# Rotational and nuclear-spin level dependent photodissociation dynamics of $H_2S$

Yarui Zhao[1,2,7], Zijie Luo[2,7], Yao Chang [2,7], Yucheng Wu[2], Su-e Zhang[2], Zhenxing Li[2], Hongbin Ding[1], Guorong Wu [2], Jyoti S. Campbell[3], Christopher S. Hansen [3✉], Stuart W. Crane[4], Colin M. Western [4], Michael N. R. Ashfold [4✉], Kaijun Yuan [2,5✉] & Xueming Yang [2,6]

The detailed features of molecular photochemistry are key to understanding chemical processes enabled by non-adiabatic transitions between potential energy surfaces. But even in a small molecule like hydrogen sulphide ($H_2S$), the influence of non-adiabatic transitions is not yet well understood. Here we report high resolution translational spectroscopy measurements of the H and S($^1D$) photoproducts formed following excitation of $H_2S$ to selected quantum levels of a Rydberg state with $^1B_1$ electronic symmetry at wavelengths λ ~ 139.1 nm, revealing rich photofragmentation dynamics. Analysis reveals formation of SH(X), SH(A), S ($^3P$) and $H_2$ co-fragments, and in the diatomic products, inverted internal state population distributions. These nuclear dynamics are rationalised in terms of vibronic and rotational dependent predissociations, with relative probabilities depending on the parent quantum level. The study suggests likely formation routes for the S atoms attributed to solar photolysis of $H_2S$ in the coma of comets like C/1995 O1 and C/2014 Q2.

[1] School of Physics, Key Laboratory of Materials Modification by Laser, Ion and Electron Beams, Chinese Ministry of Education, Dalian University of Technology, Dalian, China. [2] State Key Laboratory of Molecular Reaction Dynamics and Dalian Coherent Light Source, Dalian Institute of Chemical Physics, Chinese Academy of Sciences, Dalian, China. [3] School of Chemistry, University of New South Wales, Sydney, NSW, Australia. [4] School of Chemistry, University of Bristol, Bristol, UK. [5] University of Chinese Academy of Sciences, Beijing, China. [6] Department of Chemistry, Southern University of Science and Technology, Shenzhen, China. [7] These authors contributed equally: Yarui Zhao, Zijie Luo, Yao Chang. ✉email: christopher.hansen@unsw.edu.au; mike.ashfold@bristol.ac.uk; kjyuan@dicp.ac.cn

Sulphur is a relatively abundant element in the Universe (the S/H ratio in the solar photosphere is $\sim 1.3 \times 10^{-5}$ (refs. [1,2]) but the abundances of known sulphur-containing molecules in the interstellar medium (ISM) sum to much less than this value[3]. Estimates based on the limited range of S-containing compounds detected in low-density diffuse clouds imply sulphur fractions similar to the cosmic value[4], that decrease markedly on moving into denser regions of the ISM[5–7]. The abundances of S-containing species in the outer layers of the photodissociation region in the Horsehead nebula, for example, are thought to be only about one quarter of the cosmic value[8] and orders of magnitude lower values have been suggested in cold molecular clouds[9]. Given the high hydrogen abundances and the mobility of hydrogen in the ice matrix, sulphur atoms incident on interstellar ice mantles are expected to favour formation of $H_2S$, the chemical- and photo-induced desorption of which is considered the main source of gas-phase $H_2S$ molecules[10,11]. $H_2S$ has been detected in the atmospheres of comets P/Halley[12], C/1995 O1 (Hale-Bopp)[13,14], C/2014 Q2 (Lovejoy)[15] and 67P/Churyumov–Gerasimenko[16,17] and, where comparisons are possible, the returned $H_2S$ densities are significantly greater than those of any other sulphur-containing species. Gaseous $H_2S$ has also been detected in the Jovian atmosphere[18] and above the cloud deck in the atmospheres of Uranus[19] and (probably) Neptune[20].

The electronic spectrum of $H_2S$ displays weak continuous absorption at wavelengths $\lambda < 260$ nm and stronger absorption features at $\lambda < 155$ nm associated with excitations to Rydberg states[21–23], as illustrated in Fig. 1 and shown in more detail in Supplementary Fig. 1. Photodissociation by solar ultraviolet (UV) radiation is an important destruction route for $H_2S$ molecules in the ISM. Photolysis studies in the long-wavelength continuum[24] and at $\lambda = 157.6$ nm[25] reveal prompt S–H bond fission and formation of ground ($X^2\Pi$) state SH radicals. Lyman-$\alpha$ ($\lambda = 121.6$ nm) photolysis, in contrast, yields SH radicals in the excited $A^2\Sigma^+$ state[26,27]. These earlier data guide the current astrochemical models[28,29], which assume a very simple description of this photophysics: dissociation exclusively to H + SH fragments, supplemented by photoionization at energies above the first ionisation potential ($84432 \pm 2$ cm$^{-1}$, ref. [30], with relative probabilities determined by the respective cross-sections[31]. Recent photofragment translational spectroscopy (PTS) measurements of the H and S($^1$D) atoms from photolysis of jet-cooled $H_2S$ molecules at many wavelengths in the range $122 \leq \lambda \leq 155$ nm hint at a much richer photochemistry, however, involving multiple excited electronic states, a range of non-adiabatic inter-state couplings, and fragmentation to many of the spin-allowed dissociation limits illustrated in Fig. 1 and detailed in Table 1[32]. Of particular astrophysical significance, the PTS study showed that only $\sim 25\%$ of $H_2S$ photoexcitation events induced by the general interstellar radiation field (ISRF)[33] would yield SH(X) products; sequential fragmentation to three atoms is the most likely outcome[32]. This finding provides a plausible explanation for (i) prior rotational spectroscopy measurements directed at the W49 massive star-forming region, which deduced SH/$H_2S$ abundance ratios lower than would be predicted by the standard models in turbulent dissipation regions and shocks[34] and (ii) the detection of UV emission from S atoms attributed to $H_2S$ photodissociation in the comas of, for example, C/1995 O1(Hale-Bopp)[35] and C/2014 Q2 (Lovejoy)[36].

Inspection of Fig. 1 reveals an intense absorption feature at $\lambda \sim 139.1$ nm, which lies in the middle of the range where the branching into SH(A)/SH(X) primary photoproducts shows a strong wavelength dependence[32]. This absorption is attributable to excitation from the $\tilde{X}^1A_1$ ground state to a predissociated Rydberg state of $H_2S$ with $^1B_1$ symmetry[21,23,37–39]. The excited state has a near-integer quantum defect, encouraging assignment in terms of excitation from the highest occupied, non-bonding $3pb_1$ orbital to a $3da_1$ Rydberg orbital[21,23], and predissociates sufficiently slowly to allow excitation to specific rotational ($J_{KaKc}$) levels of the $^1B_1$ state. Earlier resonance enhanced multiphoton ionisation (REMPI) studies involving this $^1B_1$ state identified both homogeneous (i.e. vibronic) and heterogeneous (i.e. Coriolis induced) predissociation mechanisms[23,39] but were silent with regard to the products.

Here, we show the rich quantum state-dependent photofragmentation dynamics that prevail when exciting within the manifold of levels associated with just this one predissociated electronic state of $H_2S$ and serves to highlight the over-simplicity of the current astrochemistry model descriptions. The present

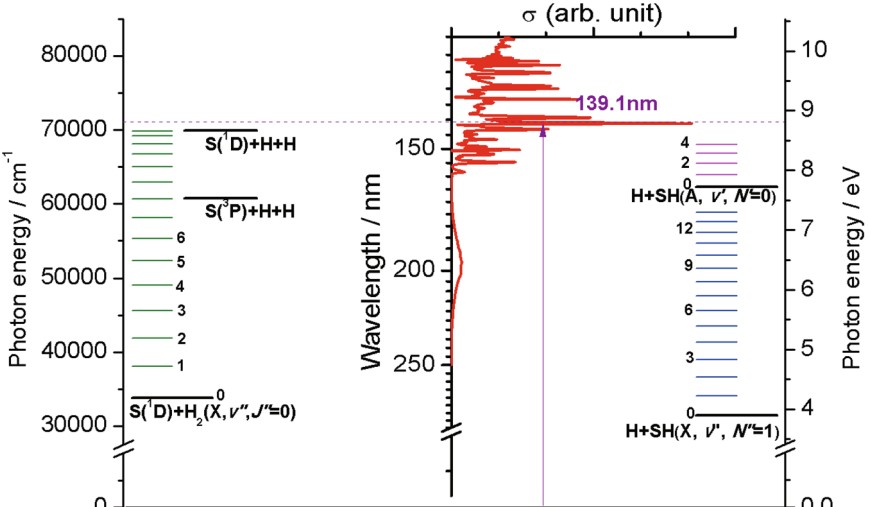

**Fig. 1 Overview of photodissociation processes of H$_2$S at $\lambda \sim 139.1$ nm.** Overview of the H$_2$S parent absorption spectrum (the red curve, plotted as cross-section $\sigma$ versus UV wavelength in nm, adapted from ref. [22]) along with the corresponding vertical excitation energies in cm$^{-1}$ and eV (left and right-hand scales, respectively) illustrating the various dissociation limits and the threshold energies for forming SH(X), SH(A) and H$_2$(X) products in their different vibrational ($v$) states (shown, respectively, by the blue, cerise and green horizontal lines). The purple arrow displays the photo-excitation process and the dashed purple line represents the energy level of 139.1 nm.

**Table 1 Thermochemical thresholds for spin-allowed fragmentation processes I–V of $H_2S$, derived using literature values for the bond dissociation energies $D_0°(HS-H)$[69], $D_0°(S-H)$[70], $D_0°(H-H)$[71] and the electronic term values $T_{00}(SH(A-X))$[72] and $\Delta E(S(^1D_2-^3P_2))$[66].**

| Process | Products | Threshold energy, $\Delta E/cm^{-1}$ |
|---|---|---|
| I | $H + SH(X^2\Pi_{3/2},\ v = 0,\ N = 1)$ | $31{,}451 \pm 4$ |
| II | $H + SH(A^2\Sigma,\ v' = 0,\ N' = 0)$ | $62{,}284 \pm 4$ |
| III | $H + H + S(^3P_2)$ | $60{,}696 \pm 25$ |
| IV | $H + H + S(^1D_2)$ | $69{,}935 \pm 25$ |
| V | $H_2(X^1\Sigma_g^+,\ v'' = 0,\ J'' = 0) + S(^1D_2)$ | $33{,}817 \pm 25$ |

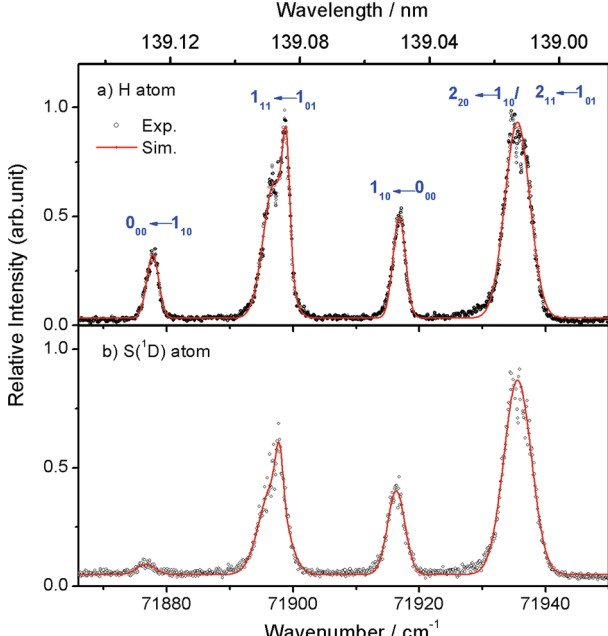

**Fig. 2 The H and S($^1$D) product photofragment excitation (PHOFEX) spectra following photoexcitation of $H_2S$. a** H and **b** S($^1$D) atom PHOFEX spectra obtained following photodissociation of a jet-cooled 30% $H_2S$ in Ar sample at wavelengths in the range $139.13 \geq \lambda \geq 138.99$ nm ($71{,}875$–$71{,}950$ cm$^{-1}$), with the dominant transitions contributing to the four features indicated. The raw data are provided as a Source Data file.

data comprise excitation spectra for forming H and S($^1$D) atoms, and $H_2$ molecules in selected vibration, rotation ($v''$, $J''$) levels, (i.e. photofragment excitation (PHOFEX) spectra) and translational energy distributions of H and S($^1$D) atom products derived using H-atom Rydberg tagging[29,40] and velocity map ion imaging[41] techniques, respectively (detailed in the 'Methods' section). These results provide a lens through which the rich photofragmentation dynamics of $H_2S$ can begin to be appreciated. Processes I–V all contribute to the decay of the photoexcited $H_2S(^1B_1)$ molecules, but with parent rotational level-dependent efficiencies. The data also confirm nuclear spin conservation in dissociation pathway V and show that some products are formed via more than one route. The details of the excited-state-resolved, multi-channel fragmentations revealed for this prototypical three-atom system are ripe for quantitative interpretation by contemporary electronic structure and excited-state dynamics studies.

## Results

**PHOFEX spectra**. Figure 2 shows excitation spectra for forming H and S($^1$D) atoms following excitation of a jet-cooled sample of $H_2S$ (30% in Ar) across the wavenumber range $71{,}865$–$71{,}950$ cm$^{-1}$ ($139.15$–$138.99$ nm). Both spectra show four features, but with clearly different relative intensities. The features can be assigned using the spectral simulation programme PGOPHER[42] and the appropriate spectroscopic parameters[23] (Supplementary Fig. 2). Each comprises one (or more) lifetime broadened transitions. The feature at $71{,}877.7$ cm$^{-1}$ is a single rovibronic transition (the $0_{00}$-$1_{10}$ line), the $71{,}916.5$ cm$^{-1}$ feature is dominated by the $1_{10}$-$0_{00}$ transition, while the major contributors to the more heavily blended features centred at ~$71{,}897$ cm$^{-1}$ and ~$71{,}936$ cm$^{-1}$ are, respectively, the $1_{11}$-$1_{01}$ and $2_{11}$-$1_{01}$/$2_{20}$-$1_{10}$ transitions. Supplementary Table 1 gives further details of the contributing transitions, the relative populations of the various ground state rotational levels at two different parent rotational temperatures ($T_{rot} = 3$ K and 15 K) and the values of $\langle J_q^2\rangle$ (the expectation values of the square of the angular momentum about the $q = a$, $b$ and $c$-inertial axes in the excited rotational level).

Several factors influence the relative intensities of these features. First, we note that these are excitation spectra for forming specific products; they report a convolution of the parent absorption probability and the branching ratio for forming the target fragment. Prior work[32] suggests that the H atom loss processes (I–IV) are dominant at these wavelengths, so the H atom PHOFEX spectrum (Fig. 2a) is likely to better approximate the parent absorption spectrum. The rate of Coriolis-driven predissociation of this $^1B_1$ excited state has previously been shown to scale with $\langle J_b^2\rangle$[23], so simulations of the $^1B_1 - \tilde{X}^1A_1$ band have to recognise both the homogeneous and heterogeneous

(*i.e.* rotational level independent and dependent) contributions to the excited state decay rate, and thus to the lifetime broadened peak linewidths and, via conservation of transition probability, the peak heights.

Symmetry dictates that each rotational level of $H_2S$ satisfies either *ortho*- or *para*-nuclear spin statistics. The former levels (for which $K_a + K_c = $ odd in the $\tilde{X}$ state) have three times higher statistical weight, and transitions involving *ortho*-$H_2S$ molecules are highlighted in bold in Supplementary Table 1a, b. *Ortho*- and *para*-$H_2S$ molecules do not interconvert during the supersonic expansion, so any simulation of the jet-cooled excitation spectra must employ different rotational ($T_{rot}$) and nuclear spin ($T_{ns}$) temperatures. Supplementary Fig. 2a, c shows H atom PHOFEX spectra recorded under different expansion conditions which, as confirmed by the accompanying PGOPHER-simulated absorption spectra (Supplementary Figs. 2b, d), afford different degrees of rotational cooling and different $T_{rot}$ values. Under the most dilute expansion conditions (yielding $T_{rot}$~3 K), almost all the parent population has relaxed to the lowest energy *para*- ($0_{00}$) and *ortho*- ($1_{01}$) levels of the ground state (see Supplementary Table 1b), the $0_{00}$-$1_{10}$ line at $71{,}877.7$ cm$^{-1}$ is no longer observed and the blended features clearly narrow.

The H and S($^1$D) atom PHOFEX spectra shown in Fig. 2 were recorded under comparable expansion conditions, wherein $T_{rot}$~15 K, yet the $0_{00}$-$1_{10}$ line is clearly much weaker and the blended ~$71{,}936$ cm$^{-1}$ feature relatively more intense in the latter. Supplementary Fig. 2e, f show, respectively, the 1 ($\lambda$~139 nm) + 1' ($\lambda = 532$ nm) parent REMPI spectrum and a PHOFEX spectrum for forming *ortho*-$H_2$ products (in the $v'' = 10$, $J'' = 1$ level). As expected, the $71{,}916.5$ cm$^{-1}$ feature associated with *para*-$H_2S$ molecules is absent in the *ortho*-$H_2$ PHOFEX spectrum (nuclear spin is conserved in the fragmentation process, as also found in studies of photoinduced $H_2$ elimination from the $H_2S^+$ parent cation[43]. But the $0_{00}$-$1_{10}$ line, which samples *ortho*-$H_2S$ molecules, is barely discernible either. These differences confirm

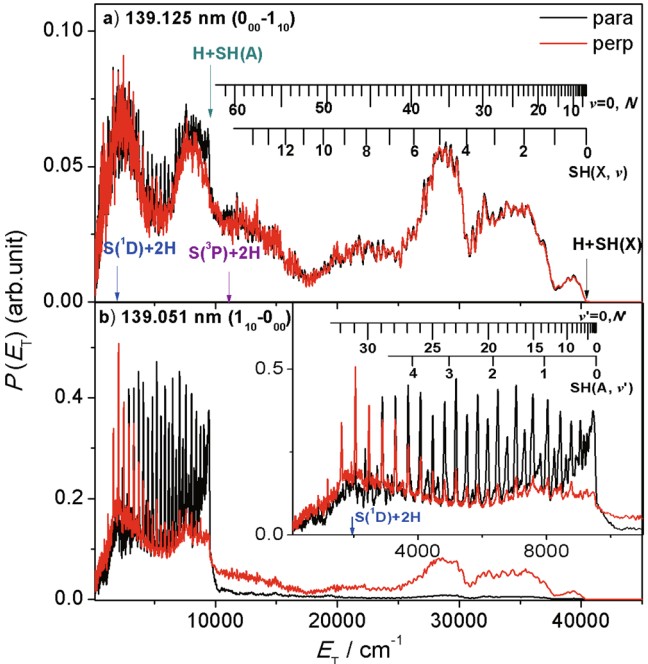

**Fig. 3 The H + SH product translational energy ($P(E_T)$) spectra from H$_2$S photodissociation.** $P(E_T)$ spectra derived from H atom TOF spectra following photodissociation of a jet-cooled 30% H$_2$S in Ar sample at $\lambda$ = **a** 139.125 and **b** 139.051 nm with $\varepsilon_{phot}$ aligned, respectively, parallel ($\theta = 0°$, black) and perpendicular ($\theta = 90°$, red) to the detection axis. The combs in **a** and in the inset to **b** show the $E_T$ values associated with formation of H atoms in conjunction with selected rovibrational levels of the primary SH (X) and SH(A) fragments, respectively. The energetic limits of these dissociation channels are marked by coloured arrows. The raw data are provided as a Source Data file.

that the predissociation rates and the branching into the various predissociation products both depend on the excited state rotational level.

**H atom product translational energy distributions.** H atom time-of-flight (TOF) spectra were recorded following photolysis of H$_2$S at wavelengths within each of the four main features in Fig. 2. As before[32], these TOF data (Supplementary Fig. 3) were converted to the corresponding total H + SH translational energy distributions, $P(E_T)$, as described in the 'Methods' section. Fig. 3 contrasts the $P(E_T)$ spectra obtained at $\lambda$ = 139.125 nm (exciting the $0_{00}-1_{10}$ transition) and 139.051 nm ($1_{10}-0_{00}$ transition), with the polarisation vector of the photolysis laser radiation ($\varepsilon_{phot}$) aligned, respectively, parallel ($\theta = 0°$) and perpendicular ($\theta = 90°$) to the detection axis. The $P(E_T)$ spectra obtained when exciting within the blended features at $\lambda$ = 139.085 and 139.015 nm are shown in Supplementary Fig. 4.

These $P(E_T)$ spectra all look rather similar at high $E_T$ (>10,000 cm$^{-1}$). The $\theta = 0°$ and 90° spectra obtained when exciting at 139.125 nm (Fig. 3a) are essentially identical (reflecting the isotropy of the $J' = 0$ wavefunction) but, at all other wavelengths, the high-$E_T$ signal is consistently more intense when detecting at $\theta = 90°$. Such recoil anisotropy is consistent with the perpendicular transition assignment ($^1B_1 - \tilde{X}^1A_1$, i.e. the transition dipole moment lies perpendicular to the molecular plane) and subsequent homogeneous predissociation on a timescale (~picosecond) that is shorter than the parent rotational period.

Given the threshold energies for the various fragmentation paths (Table 1 and illustrated also in Fig. 1), the structured envelope with

$E_T > 11,000$ cm$^{-1}$ must be associated with SH(X) co-fragments formed via process I in a spread of vibrational ($v''$) and rotational ($N''$) quantum states. The best-fit simulation of this spectrum is shown in Supplementary Fig. 5. The higher energy part (with $E_T > 18,000$ cm$^{-1}$) is attributable to formation of SH(X) fragments with $0 \le v'' \le 10$ and a spread of low $N''$ values (peaking at $N''{\sim}10$). These SH(X, $v''$, $N''$) population distributions have similarities with those reported when exciting H$_2$S at $\lambda = 157.6$ nm[25]. The lower energy part, with $11000 < E_T < 18,000$ cm$^{-1}$, spanning beyond the three-body dissociation limit to H + H + S($^3$P) atoms (process III), is attributable to formation of H + SH(X, low $v''$, high $N''$) products with energies extending beyond the SH(X) state bond dissociation energy, $D_0$(S–H). The broad maximum in Fig. 3a centred at $E_T{\sim}8000$ cm$^{-1}$ is likely to include contributions from H atoms formed with primary SH(X) fragments in 'super-rotor' levels, i.e. quasi-bound levels supported by the accompanying centrifugal potential energy barrier. Population of analogous OH(X) super-rotor levels in the photolysis of H$_2$O has been reported[44]. Many of these SH(X) super-rotors will predissociate by tunnelling through the centrifugal barrier within the short (<5 ns) time delay between the photolysis and probe (H Rydberg tagging) laser pulses to yield a second (slow) H atom. The broad peak centred at $E_T \sim 2500$ cm$^{-1}$ in Fig. 3a is attributed to such secondary H atom products.

The spectra obtained at wavelengths that sample $^1B_1$ state levels with $J' > 0$ show another structured component at $E_T \le 10,000$ cm$^{-1}$. To highlight these features, the distribution shown in Supplementary Fig. 5a has been used as a basis function ($P(E_T)_{vib}$) that represents the contribution from homogeneous (i.e. purely vibronic) predissociation pathways and a suitably weighted amount of this $P(E_T)_{vib}$ function subtracted from the $P(E_T)$ distributions obtained at $\lambda = 139.085$, 139.051 and 139.015 nm, so as to minimise the signal at $E_T > 10,000$ cm$^{-1}$. The resulting $P(E_T)_{Cor}$ distributions (Supplementary Fig. 6) describe the Coriolis-induced predissociation yields and confirm formation of H + SH(A) products (process II).

These SH(A) fragments are mainly formed in the $v' = 0$ level, in a broad spread of rotational ($N'$) levels extending to (and just beyond) the SH(A) state bond dissociation energy (i.e. to energies above the threshold for forming H + S($^1$D) atoms)—as shown by the comb included in Fig. 3b[32] – and with an $N$-dependent recoil anisotropy: H + SH(A, $v' = 0$, low $N'$) products recoil preferentially along the axis parallel to $\varepsilon_{phot}$, whereas H + SH(A, $v' = 0$, high $N'$) products appear with greater probability along axes perpendicular to $\varepsilon_{phot}$.

All SH(A) radicals predissociate on a nanosecond (or shorter) timescale to yield H + S($^3$P$_J$) atom products[45]. Thus the primary SH(A) photoproducts revealed in Fig. 3b and Supplementary Fig. 4 must decay to yield a second H atom within the time that the Rydberg tagging laser radiation is present, and these secondary H atoms must also contribute to the $P(E_T)_{Cor}$ distribution and the total $P(E_T)$ spectra. The predissociation of SH(A) radicals favours population of ground ($J = 2$) spin-orbit state S($^3$P$_J$) products[45], and combs indicating the $E_T$ values of H + S($^3$P$_2$) products expected from predissociation of selected SH (A, $v' = 0$, $N'$) photoproducts are also included in Supplementary Fig. 6. Astute readers will recognise weak structure attributable to H + SH(A) products in Fig. 3a. This is attributed to dissociation following excitation to the weak absorption continuum that underlies the $^1B_1 - \tilde{X}^1A_1$ band, since similar signal is also evident in $P(E_T)$ spectra obtained when exciting at wavelengths off-resonant with the $0_{00}-1_{10}$ transition (e.g. at $\lambda = 139.117$ nm, Supplementary Fig. 7).

In summary, the H Rydberg atom photofragment translational spectroscopy (HRA-PTS) measurements reveal formation of (i) H + SH(X) products via vibronic predissociation from the $^1B_1$

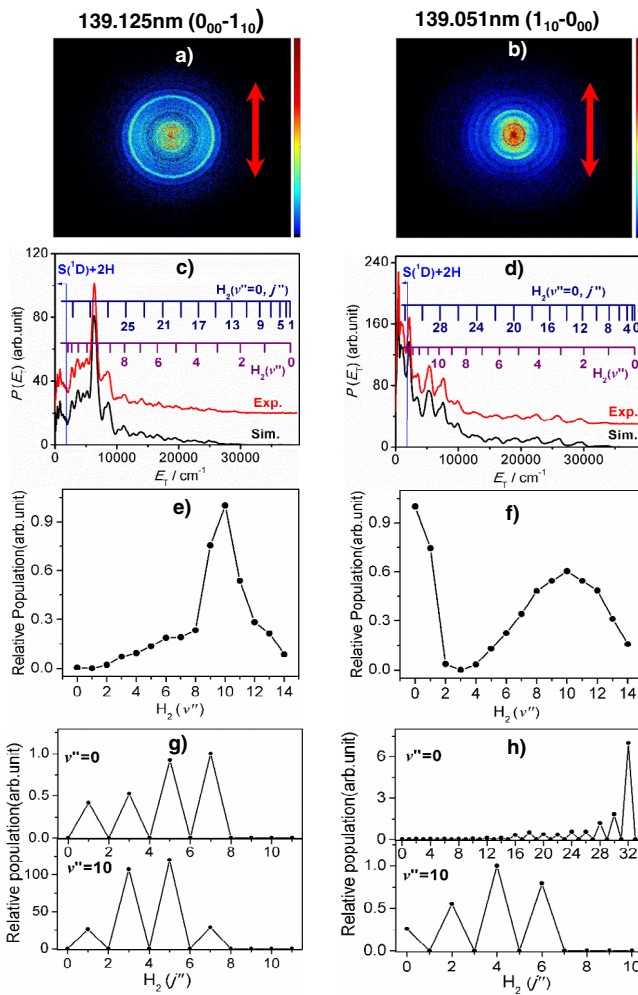

**Fig. 4 The S($^1$D) + H$_2$ product translational energy distributions and the H$_2$ product quantum state population distributions.** Time-sliced velocity map images of the S($^1$D$_2$) photofragments from photolysis of H$_2$S at $\lambda$ = **a** 139.125 and **b** 139.051 nm with $\varepsilon_{phot}$ aligned vertically in the plane of the image (as illustrated by the double-headed red arrow). The $P(E_T)$ spectra derived from these images are shown in **c** and **d**, in red, along with the best-fit simulations of the spectra at $E_T$ values down to the S($^1$D) + H$_2$ dissociation threshold, in black and offset vertically for clarity. The superposed combs in **c** and **d** indicate the $E_T$ values associated with formation of the various H$_2$($v''$, $J''$ = 0) and, respectively, the *ortho-* and *para-*$J''$ states of H$_2$($v''$ = 0). The energetic limit of the S($^1$D) + 2H channel is marked by the vertical blue line. **e** and **f** show the H$_2$($v''$) state population distributions returned by the respective best-fit simulations, while **g** and **h** show the $J''$ state population distributions for H$_2$ products in representative low ($v''$ = 0) and high ($v''$ = 10) vibrational states formed at each wavelength. The raw data are provided as a Source Data file.

Rydberg state, (ii) H + SH(A) products via Coriolis-induced predissociation of excited state levels with $<J_b^2>> 0$ and (iii) S($^3$P) atoms via sequential decay of primary SH(X) and SH(A) photoproducts (*i.e.* net process III, by tunnelling from 'super-rotor' levels and by electronic predissociation, respectively).

**S($^1$D) atom product translational energy distributions**. Figure 4a, b shows time-sliced velocity map images of the S($^1$D) photofragments measured following photolysis of H$_2$S at $\lambda$ = 139.125 and 139.051 nm, respectively, and subsequent resonant ionisation at $\lambda$ = 130.092 nm. Image analysis yields the

corresponding $P(E_T)$ distributions (Fig. 4c, d), derived assuming momentum conservation and H$_2$ as the partner fragment.

The structure in these $P(E_T)$ spectra reveals population of specific H$_2$($v''$, $J''$) levels, which are clearly very different in the two cases. Best-fit simulations of these spectra (Fig. 4c, d) return the H$_2$($v''$) population distributions shown in Fig. 4e, f. Rotational ($J''$) distributions for representative $v''$ = 0 and $v''$ = 10 state products are shown in Fig. 4g, h. The H$_2$ products formed when exciting the $0_{00}$–$1_{10}$ transition show an isotropic velocity distribution, and an inverted vibrational state population distribution spanning all bound vibrational levels of H$_2$ (i.e. all levels with $v'' \leq 14$), peaking at $v''$ = 10, but only modest rotational excitation. The H$_2$ products formed from the $1_{10}$–$0_{00}$ transition, in contrast, have anisotropic velocity distributions and a bimodal vibrational state population distribution comprising not just an inverted vibrationally 'hot', rotationally 'cold' component (again peaking at $v''$~10) but also a substantial yield of vibrationally 'cold' (*i.e.* $v''$ = 0 and 1) products. These H$_2$(low $v''$) products show highly inverted rotational state population distributions; the $P(E_T)$ spectrum shows structure at low $E_T$ values, beyond the S($^1$D) + 2H dissociation limit, consistent with formation of super-rotor levels of H$_2$($v''$ = 0). Fig. 4g, h) confirm the expected conservation of nuclear spin symmetry: the H$_2$ products from excitation of (a) *ortho-* and (b) *para-*H$_2$S molecules are formed in, respectively, odd and even $J''$ levels.

Supplementary Fig. 8 shows images obtained following excitation at $\lambda$ = 139.085 and 139.015 nm, along with the derived $P(E_T)$ and H$_2$ internal energy distributions. The H$_2$($v''$, $J''$) distributions measured at $\lambda$ = 139.085 nm are reminiscent of those found when exciting to the $0_{00}$ level (Fig. 4e, g); excitation at this wavelength populates primarily the $1_{11}$ level (another level of *ortho-*H$_2$S and the only other excited level for which $<J_b^2> = 0$). The H$_2$($v''$, $J''$) distributions derived from the image recorded at $\lambda$ = 139.015 nm, in contrast, are more like those found when exciting to the $1_{10}$ level (Fig. 4f, h). Again, the dominant excitations at this wavelength populate levels with $<J_b^2>> 0$, though these levels ($2_{11}$ and $2_{20}$) are levels of *ortho-*H$_2$S and the H$_2$ products thus have odd $J''$. Supplementary Fig. 9 shows S($^1$D) images recorded at other wavelengths near $\lambda$~139.09 nm, which illustrate the extreme sensitivity of the product energy disposal to the exact choice of excitation wavelength within these blended features.

These ion imaging studies reveal formation of S($^1$D) + H$_2$ products (*i.e.* process V) via both vibronic and Coriolis-induced predissociation pathways. The H$_2$ products arising via the former route carry substantial vibrational but little rotational excitation. Previous REMPI studies have reported formation of H$_2$ products in high $v''$, low $J''$ states following excitation of H$_2$S to similar total energies[46–50]. The $P(E_T)$ distributions measured when exciting parent levels with $<J_b^2>> 0$ reveal a rival Coriolis-induced pathway yielding additional H$_2$ products, characterised by little vibrational but massive rotational excitation—including population of super-rotor levels of H$_2$.

**Discussion**

The present study affords detailed views of different photofragmentation pathways in a prototypical triatomic molecule. Photoexcitation to the predissociated $^1$B$_1$ state of H$_2$S at $\lambda$~139.1 nm allows definition of the initial rovibrational level(s) from which dissociation occurs (i.e. their rotational angular momentum and nuclear spin symmetry), while the excitation and PTS detection methods yield quantum-state-resolved information on the dissociation products. Qualitatively, the deduced dynamics can all be reconciled within the framework illustrated in Fig. 5, but a complete interpretation will require much better knowledge

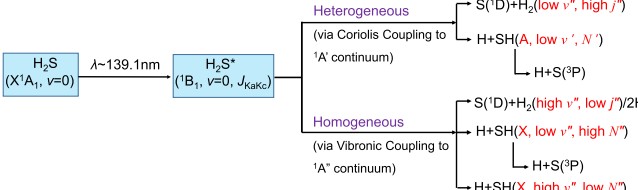

**Fig. 5 Photodissociation mechanisms of $H_2S$ upon excitation at $\lambda \sim 139.1$ nm.** Schematic illustration of the deduced heterogeneous (Coriolis-induced) and homogeneous (vibronic coupling) predissociation pathways following excitation to the $^1B_1$ Rydberg state of $H_2S$ at $\lambda \sim 139.1$ nm. $H_2S^*$ represents the energetic molecule formed by absorbing one 139.1 nm photon.

of the topographies of, and non-adiabatic couplings between, the various excited-state potential energy surfaces (PESs). Hopefully, the scope of the present data will inspire state-of-the-art computational studies of $H_2S$, enabling it to acquire status as a reference molecule within the photophysics and chemical reaction dynamics communities—comparable to that enjoyed by its lighter analogue $H_2O$[51].

Previous REMPI[52] and OH(A) PHOFEX[53] spectroscopy and HRA-PTS studies[54] following excitation to the analogous $\tilde{C}^1B_1$ state of $H_2O$ have also revealed competitive homogeneous (vibronic) and heterogeneous (Coriolis-induced) predissociation pathways. These have been rationalised by non-adiabatic coupling from the $\tilde{C}$ state to dissociative states of, respectively, $^1B_1$ and $^1A_1$ symmetry (labelled the $\tilde{A}$ and $\tilde{B}$ states of $H_2O$)[51] and, from hereon, it proves helpful to discuss the fragmentation dynamics of $H_2S$ revealed by the present work in the context of existing knowledge relating to the fragmentation of $H_2O$.

**Vibronically-induced predissociation.** The topography of the $\tilde{A}^1B_1$ PES of $H_2O$ ensures that direct population of this state by vertical photoexcitation from the ground state equilibrium geometry leads to prompt O–H bond fission, yielding $H + OH(X)$ fragments with modest rovibrational excitation of the latter[55–57]. The $\tilde{C}$ and $\tilde{X}$ states of $H_2O$ have similar equilibrium geometries, yet vibronic predissociation from the $\tilde{C}$, $v = 0$, $0_{00}$ level yields OH (X) products in a wide range of vibrational levels ($0 \leq v'' \leq 13$). Wavepacket calculations[58] provide an explanation for this striking energy disposal: non-adiabatic coupling between the $\tilde{C}$ and $\tilde{A}$ state PESs is mediated by sampling an intermediate $^1A_2$ state PES[59], most efficiently at compressed O–H bond lengths, and this compression of the surviving O–H bond maps into the final product vibration.

The present data for $H_2S$ show obvious parallels, but also some differences. Additionally, the present data inform on the competing $H_2$ elimination channel—the possible $O(^1D) + H_2$ product channel following VUV photoexcitation of $H_2O$ has yet to be studied in any detail. Prior studies of $H_2S$ photoexcitation within its long-wavelength absorption band (see Fig. 1), the analogue of the $\tilde{A}$–$\tilde{X}$ absorption of $H_2O$, reveal the first important difference: Vertical excitation of $H_2S$ samples not one but two near degenerate excited states (with $^1B_1$ and $^1A_2$ symmetry in $C_{2v}$, i.e. both $^1A''$ in $C_s$), only one of which is dissociative upon H–SH bond extension[60]. The increasing vibrational excitation of the SH (X) products observed when exciting at shorter wavelengths within this continuum[25] has been attributed to efficient electronic predissociation of molecules initially excited to the higher-lying, quasi-bound state[61,62].

Thus, the analogue of the $\tilde{A}$ state of $H_2O$ in $H_2S$ is probably better viewed as a 'lumpy continuum' of quasi-bound resonances embedded in a dissociative continuum, all with $^1A''$ symmetry,

appropriate for accession by vibronic (i.e. rotation-independent) coupling from the $^1B_1$ state of current interest. High-level ab initio calculations of these coupled excited state PESs and accompanying wavepacket propagations would likely reveal whether the foregoing explanations (i.e. compression of the S–H bonds at the point of optimal non-adiabatic coupling to the $^1A''$ continuum, and/or a legacy of the resonance structure within the $^1A''$ continuum) can account for the observed vibrationally excited SH(X) fragments from vibronic predissociation of $H_2S$ ($^1B_1$) molecules. These same calculations should also inform on the dynamics of $S(^1D) + H_2(X$, high $v''$, low $J'')$ product formation after coupling to the $^1A''$ continuum, which has been predicted to be a barrierless process at near-$C_{2v}$ (i.e. T-shaped) geometries[63]. We return later to consider potential sources of the $H + SH(X$, low $v''$, very high $N'')$ and $S(^3P) + 2H$ products via a vibronic coupling mechanism (as implied by the data shown in Fig. 3 and Supplementary Fig. 5).

**Coriolis-induced predissociation.** Analogy with $H_2O$ again provides a useful starting point. Vertical excitation to the $\tilde{B}^1A_1$ state of $H_2O$ also results in prompt dissociation, but the O–H bond extension occurs in tandem with rapid opening of the $\angle$HOH bond angle: some of the photoexcited molecules evolve on the adiabatic $\tilde{B}$ state PES and undergo H–OH bond fission to yield electronically excited OH(A) fragments with little vibrational but very high rotational excitation—the legacy of the strong angular forces imposed by the topography of the $\tilde{B}$ state PES. However, the dominant decay processes for $H_2O(\tilde{B})$ molecules involve non-adiabatic couplings (i) via a Renner–Teller seam of degeneracy between the $\tilde{B}$ and $\tilde{A}$ state PESs at linear geometries and (ii) at either of two conical intersections (CIs) between the $\tilde{B}$ and $\tilde{X}$ state PESs at linear H….OH and H….HO configurations—all of which yield OH(X, low $v''$, high $N'')$ products[64,65]. $H_2O$ molecules accessing the $\tilde{B}$ state PES by Coriolis-enabled predissociation from rovibrational levels of the $\tilde{C}^1B_1$ state[54] show similar propensities for forming both OH(A) and OH(X) photofragments in low $v$, high $N$ quantum states.

Figure 3 confirms formation of $H + SH(A$, low $v'$, high $N')$ products by rotationally-mediated predissociation from $H_2S(^1B_1)$ state levels with $\langle J_b^2 \rangle >> 0$. Rotation about the $b$ axis (the $z$ inertial axis in $C_{2v}$) transforms as $a_2$ and can thus promote non-adiabatic coupling of the $^1B_1$ state to a continuum of $^1B_2$ symmetry. This highlights another difference cf. $H_2O$. The above discussion of the long-wavelength absorption suggests that, in $H_2S$, the analogue of the $\tilde{B}$ state of $H_2O$ will also comprise two nested states, of $^1A_1$ and $^1B_2$ symmetry in $C_{2v}$ (i.e. both $^1A'$ in $C_s$). These are the upper components of the Renner–Teller pairs with, respectively, the lower-lying $^1B_1$ and $^1A_2$ states. Both $^1A'$ states will have linear minimum energy geometries at short $R_{H-SH}$ bond lengths and the coupled states should manifest as a series of quasi-bound resonances embedded in a continuum that correlates adiabatically to the $H + SH(A)$ asymptote. The topography of the dissociative $^1A'$ PES will encourage H–SH bond extension in concert with opening of $\angle$HSH, consistent with the observed $H + SH(A$, low $v'$, high $N')$ products.

The imaging data (Fig. 4b) reveal that formation of $S(^1D) + H_2(\text{low } v''$, high $J'')$ products also requires Coriolis-coupling to the $^1A'$ continuum and that these products recoil preferentially parallel to $\varepsilon_{phot}$. Both observations can be explained assuming a set of dissociative trajectories on the $^1A'$ PES for which the balance of axial and tangential forces allows the molecule to evolve outside the region of the CI at linear H….SH geometries (CI-1 in Supplementary Fig. 10) and thus remain on the $^1A'$ PES but not dissociate fully. Rather, the emerging H and SH(A) fragments are held in a centrifugally-bound complex and are

drawn into a seam of intersection between the $^1A'$ and $\tilde{X}$ state PESs at linear H….HS geometries. This seam, which includes the CI-2 depicted in Supplementary Fig. 10 but spans a wide range of H…H and S…H separations, enables H atom transfer and ultimate formation of the observed S($^1D$) atoms and $H_2$ fragments. The experimental data show that this fragmentation route favours massive rotation of the $H_2$ products and an extension of such dynamics could contribute to the observed S($^1D$) + 2H fragment yield.

**Another vibronic predissociation mechanism.** We now consider possible routes to the observed H + SH(X, low $v''$, high $N''$) and S($^3P$) + 2H products. Many VUV photolysis studies of $H_2O$ have identified H + OH(X, low $v''$, high $N''$) products but, in all cases, the high product rotation is seen as a legacy of initial motion (opening of $\angle$HOH) on the $\tilde{B}$ state PES prior to non-adiabatic coupling to the $\tilde{A}$ or $\tilde{X}$ state PESs. But the H + SH(X, low $v''$, high $N''$) products revealed in Fig. 3 are deduced to arise via a vibronic predissociation mechanism. As at shorter excitation wavelengths[26,27,32], non-adiabatic coupling to the $^1A'$ PES provides an efficient route to H + SH(A, low $v'$, high $N'$) products, but not to H + SH(X, low $v''$, high $N''$) products—probably because the balance of angular and radial forces prevailing on the $^1A'$ PES carry all dissociating molecules outside the region of configuration space that supports non-adiabatic transfer to the lower energy states (i.e. preclude type I trajectories whereby excited molecules achieve linearity at shorter $R_{H-SH}$ values than CI-1 in Supplementary Fig. 10)[32]. But it is hard to envisage any route to very highly rotationally excited SH fragments that do not depend on the angular acceleration provided by the topography of the $^1A'$ PES.

Detailed understanding must await future high-level theoretical studies, but we can suggest a possible rotation-independent mechanism. The photoexcited $^1B_1$ molecules undergo vibronic predissociation to the $^1A''$ 'lumpy continuum'. Some survive in quasi-bound bending levels long enough to sample a broader range of configuration space, including near-linear geometries that enable non-adiabatic coupling to the upper ($^1A'$) Renner–Teller components. Further angular acceleration is then generated by passage through CI-1 between the $^1A'$ PES and the $\tilde{X}$ state PES at linear H….SH geometries (Supplementary Fig. 10), ultimately yielding H + SH(X, low $v''$, high $N''$) products – as observed. The non-observation of H + SH(A) fragments via this vibronic predissociation route can be understood by recognising that the execution of this more tortuous route to the $^1A'$ PES partitions sufficient energy into other degrees of freedom to 'close off' the excited product asymptote.

**Product branching.** Recent PTS studies showed the progressive switch from single S–H bond fission (process I), which dominates at $\lambda > 150$ nm, to three body dissociation at shorter photolysis wavelengths[32]. The predissociated $^1B_1$-$\tilde{X}^1A_1$ band investigated here falls at a wavelength where processes I–V all contribute to the product yield, via a range of non-adiabatic coupling pathways, with excited rotational level-dependent efficiencies. Only two excited state rotational levels have $\langle J_b^2 \rangle = 0$ (the $0_{00}$ and $1_{11}$ levels), so higher temperature $H_2S$ samples will contain a larger fraction of molecules with $\langle J_b^2 \rangle >> 0$ and higher average $\langle J_b^2 \rangle$ values—both of which will increase the probability of heterogeneous predissociation. Using PGOPHER[42], along with the previous parameterisation of the vibronic and Coriolis-induced predissociation rates from the $^1B_1$ state[23], suggests that the cross-section for absorption that results in heterogeneous predissociation ($\sigma_{hetero}$) contributes only ~17% of the total $^1B_1-\tilde{X}^1A_1$ cross-

section ($\sigma_{tot}$) at temperatures $T \leq 30$ K but starts to dominate once $T = 300$ K, where $\sigma_{hetero}/\sigma_{tot}\sim0.52$.

Increasing the relative probability of coupling to the $^1A'$ continuum can be expected to reduce the relative yield of H + SH (X) products, but more quantitative discussions would also require better knowledge of how the rotational angular momentum of the photoexcited molecule may continue to influence the nuclear motions after initial non-adiabatic coupling to the $^1A'$ or $^1A''$ continua. Qualitatively, however, the current data suggest that the initial non-adiabatic coupling has a major influence on the eventual product branching. For example, the decomposition of $P(E_T)$ spectra when exciting the four main features (Fig. 2) and detecting along an axis at $\theta = 54.7°$ (the magic angle variants of the spectra shown in Fig. 3 and Supplementary Fig. 4) shows the $P(E_T)_{Cor}/P(E_T)_{vib}$ ratio increasing from 0 at $\lambda = 139.125$ nm (when exciting the $0_{00}$ level) to >1 at $\lambda = 139.015$ nm (when exciting levels with $\langle J_b^2 \rangle$ in the range 2.5–4). Similarly, if we attribute the S($^1D$) + $H_2(v'' = 0$ and 1, high $J''$) products observed at $\lambda = 139.051$ and 139.015 nm to the Coriolis-induced predissociation pathway, and all other S($^1D$) + $H_2$(high $v''$, low $J''$) products to vibronic predissociation, then the ratio of the heterogeneous to homogeneous contributions to the total S($^1D$) + $H_2$ yield increases from 0 at $\lambda = 139.125$ nm to ~0.5 at $\lambda = 139.015$ nm.

In summary, this work provides one of the most complete experimental studies of molecular photofragmentation processes reported to date, affording initial parent quantum state selection and detailed investigation of competing product channels. Predissociation of the $^1B_1$ Rydberg state of $H_2S$ populated by photoexcitation at $\lambda \sim 139.1$ nm is initiated via both rotation-free (vibronic) and rotation-induced non-adiabatic couplings, thus ensuring that the relative yields of H, S, SH and $H_2$ products, their velocity distributions and respective quantum state population distributions are sensitively dependent on the chosen parent quantum state. The fragmentation mechanisms are rationalised based on available knowledge regarding the topographies of, and non-adiabatic couplings between, the PESs of the lower-lying valence excited states of $H_2S$. Similarities and differences with the photofragmentation dynamics of the more thoroughly studied homologue, $H_2O$, are highlighted. The time is now ripe for a thorough investigation of the excited state photophysics of $H_2S$ combining cutting-edge experiments of the types described here and state-of-the-art quantum chemistry methods.

From an astrophysical perspective, recent studies have shown that $H_2S$ photoexcitation by the general ISRF should favour triple fragmentation to the constituent atoms (the dominant process at wavelengths $\lambda < 130$ nm) over binary dissociation to H + SH(X) radicals (which dominates at $\lambda > 150$ nm) by a factor of ~3:1[32]. The present study explores the fate of $H_2S$ molecules excited on a strong absorption feature in the intermediate wavelength region, where the branching between two- and three-body dissociation is changing rapidly with wavelength, and reveals that this branching is also highly parent quantum state dependent. The $^1B_1$ state molecules prepared with $\langle J_b^2 \rangle = 0$ dissociate predominantly to H + SH(X) products, whereas molecules with $\langle J_b^2 \rangle > 0$ can also decay to H + SH(A) (and thence to H +H + S($^3P$)) and S($^1D$) + $H_2$ products, with probabilities that scale with $\langle J_b^2 \rangle$. Thus, the fate of $H_2S$ molecules following excitation on the strong $^1B_1-\tilde{X}^1A_1$ absorption (and, most probably, on any of the other neighbouring Rydberg resonances evident in Fig. 1) will be sensitive to the local temperature. Any S($^1D$) photoproducts formed in rarefied interstellar environments will decay radiatively (via the spin-forbidden $^1D\rightarrow^3P$ transition, with Einstein A-coefficient $A \sim 2.1 \times 10^{-2}$ s$^{-1}$ (ref. [66])), adding to the S($^3P$) yield from the various three-body fragmentation pathways of $H_2S$ and the

secondary photolysis of primary SH(X) fragments that are predicted[15] to be the main sources of the S atoms detected in the coma of several comets[35,36].

## Methods

**The H atom product translational energy distributions**. The H atom product translational energy distributions were recorded using a tuneable VUV pump source along with the H-atom Rydberg tagging time-of-flight (HRTOF) probe technique[67]. In the HRTOF detection method, the H atom products were excited from the ground state to a high $n$ Rydberg state via a two-step excitation. Step one involves resonant excitation from the $n = 1$ to $n = 2$ state at the Lyman-$\alpha$ wavelength ($\lambda = 121.6$ nm), while step two uses UV laser excitation at $\lambda \sim 365$ nm to further excite the H atom from the $n = 2$ state to a high-$n$ ($n = 30$–$80$) Rydberg state, lying slightly below the ionisation threshold. Coherent 121.6 nm radiation was generated by difference four-wave mixing (DFWM) involving two 212.556 nm photons and one 845 nm photon overlapped in a stainless steel cell filled with a 3:1 ratio Ar/Kr gas mixture. Laser light at $\lambda = 212.556$ nm was produced by doubling the output of a 355 nm (Nd:YAG laser, Spectra Physics Pro-290) pumped dye laser (Sirah, PESC-G-24) operating at $\lambda \sim 425$ nm. Half of the 532 nm output of the same Nd:YAG laser was used to pump another dye laser (Continuum ND6000) which operated at $\lambda \sim 845$ nm. The $\lambda \sim 365$ nm laser radiation used in the second step of the H-atom Rydberg tagging was generated by doubling the output of a third dye laser (Radiant Dye Laser-Jaguar, D90MA) operating at $\lambda \sim 730$ nm, which was pumped by the remaining 532 nm output of the Nd:YAG laser. To eliminate background signals arising from $\lambda = 212.556$ nm photolysis of $H_2S$ in the interaction region, the 121.6, 212.556 and 845 nm beams were passed through a biconvex LiF lens positioned off-axis at the exit of the Ar/Kr gas cell thereby ensuring that only the VUV beam was dispersed through the interaction region.

The tuneable VUV photolysis source for $H_2S$ photodissociation at $\lambda \sim 139.1$ nm was also generated by DFWM using 212.556 nm photons and tuneable radiation with $\lambda \sim 450$ nm in another mixing cell, which was coupled to the other side of the main chamber. The 212.556 nm and 450 nm photons were generated using the second Nd:YAG laser to pump two further dye lasers, respectively. The same dispersion strategy employing an off-axis mounted LiF lens was also used to ensure that the 139.1 nm (but not the 212.556 nm) radiation passed through the interaction region. Since 121.6 nm photons also induce $H_2S$ photolysis and thus generate H atom signals, it was necessary to use a background subtraction method, whereby the $\lambda \sim 139.1$ nm photolysis laser was alternated on and off. The parallel ($\theta = 0^o$) and perpendicular ($\theta = 90^o$) signals were recorded by tuning the polarisation of the 139.1 nm radiation, using a rotatable half-waveplate to rotate the polarisation of the tuneable ($\lambda \sim 450$ nm) radiation.

The neutral Rydberg-tagged H atom photofragments flew a known distance $d$ ($\sim 280$ mm) before reaching a grounded mesh-mounted close in front of Z-stack micro-channel plate (MCP) detector, where they were field-ionised immediately by the $\sim 2000$ V cm$^{-1}$ electric field. The signal detected by the MCP was then amplified by a fast pre-amplifier and counted by a multichannel scaler. The recorded TOF data (shown in Supplementary Fig. 3) were converted to the corresponding H atom kinetic energy distributions. Momentum conservation arguments were then used to derive the total translational energy distributions $P(E_T)$, where

$$E_T = \frac{1}{2}m_H\left(\frac{d}{t}\right)^2 (1 + m_H/m_{SH}), \tag{1}$$

$m$ is the photofragment mass, $d$ is the flight distance and $t$ is the TOF measured over this distance. The H atom PHOFEX spectra were recorded by integrating the H atom signals while scanning through a range of photolysis wavelengths.

**The S($^1$D) atom product translational energy distributions**. The S($^1$D) atom product translational energy distributions were recorded using the VUV pump—time-sliced velocity map imaging (TSVMI) probe technique[32]. Briefly, the pulsed supersonic beam was generated by expanding a mixture of 30% $H_2S$ and Ar into the source chamber where it was skimmed before entering (through a 2 mm hole in the first electrode), and propagating along the centre axis of, the ion optics assembly mounted in the reaction chamber. The molecular beam was intersected at right angles by the photolysis and probe laser beams between the second and the third plates of the ion optics assembly. The $\lambda \sim 139$ nm photolysis photons were generated by DFWM, as described above, with $\varepsilon_{phot}$ fixed in the horizontal plane and thus parallel to the front face of the MCP detector. The S($^1D_2$) photoproducts were probed by one photon excitation at $\lambda = 130.092$ nm, which populates the autoionizing $3p^3(^2D^o)5s$; $^1D_2^o$ level. These photons were also generated by DFWM, by combining the frequency doubled output from one dye laser (at $\lambda = 212.556$ nm) with the fundamental output of a second dye laser (at $\lambda = 580.654$ nm) in a Kr/Ar gas mixture. The resulting $S^+$ ions are accelerated through the remaining ion optics and travel through a 740 mm long field-free region before impacting on a 70 mm-diameter chevron double MCP detector coupled with a P43 phosphor screen. Transient images on the phosphor screen were recorded by a charge-coupled device camera, using a 30 ns gate voltage pulse to acquire time-sliced images.

**Parent REMPI spectra and $H_2(v'', J'')$ PHOFEX spectra**. Parent REMPI spectra and $H_2(v'', J'')$ PHOFEX spectra were recorded at UNSW Sydney using an ion-imaging apparatus[68]. Briefly, a molecular beam of 10% $H_2S$ in helium was generated using a pulsed valve (General Valve Series 9, 0.5 mm orifice, controlled by an Iota One valve driver) and passed through a 1 mm diameter skimmer into an ion-imaging spectrometer configured in spatial mapping mode. The early-time component of the molecular beam was intersected with a $\lambda \sim 139.1$ nm laser pulse generated by DFWM in a stainless-steel gas cell charged with 5 mbar of krypton. The $\lambda = 212.556$ nm and $\lambda \sim 450$ nm precursor laser pulses were generated using two different Sirah Cobra-Stretch dye lasers ($2 \times 1800$ g/mm grating). The $\lambda = 212.556$ nm radiation was generated by frequency-tripling the output of one dye laser running DCM dye in ethanol and pumped with the Nd:YAG 2nd harmonic (532 nm). The $\lambda \sim 450$ nm light was the fundamental output of the other dye laser pumped by the Nd:YAG 3rd harmonic (355 nm) and operating with Coumarin-450 laser dye. To record REMPI spectra of the $H_2S$ parent molecule, the $\lambda \sim 139.1$ nm pulse and final probe (ionisation) laser pulse, provided by the 2nd harmonic (532 nm) of a Nd:YAG laser (Quantel Brilliant B), were overlapped in space and time. The (multiphoton) probe pulse for recording $H_2$ PHOFEX spectra was provided by the doubled output ($\lambda \sim 293$ nm) of a 532 nm pumped Lambda-Physik LPD3000 dye laser running Pyromethene 597 dye and, in this case, the probe pulse was delayed by $\sim 40$ ns relative to the parent pump pulse. All wavelengths were verified using a Toptica WS5 wavelength-metre.

## Data availability

The source data underlying Figs. 2, 3 and 4 are provided as a Source Data file. All other data supporting this study are available from the authors upon request. Source data are provided with this paper.

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

## Acknowledgements

The work in Dalian was supported by the Strategic Priority Research Program of the Chinese Academy of Sciences (Grant No. XDB17000000), the Chemical Dynamics Research Center (Grant No. 21688102), the National Natural Science Foundation of China (NSFC Nos. 21873099 and 21922306), the Key Technology Team of the Chinese Academy of Sciences (Grant No. GJJSTD20190002), the international partnership programme of Chinese Academy of Sciences (No. 121421KYSB20170012), and Liaoning

Revitalization Talents Program (Grant Nos. XLYC1907154). C.S.H. is grateful for an Australian Research Council Discovery Early Career Award (DE200100549), M.N.R.A. is grateful for funding from the UK Engineering and Physical Sciences Research Council (EPSRC, EP/L005913) and both C.S.H. and M.N.R.A. are grateful to the NSFC Center for Chemical Dynamics for the award of Visiting Fellowships. We thank the DCLS staff for technical support.

## Author contributions

K.J.Y., C.S.H. and M.N.R.A. designed the experiments. Y.R.Z., Z.J.L., Y.C., Y.C.W., S.Z., Z.X.L., J.S.C., C.S.H. and K.J.Y. performed the experiments. K.J.Y., M.N.R.A., S.W.C., C. M.W., Y.R.Z., Z.J.L. and C.S.H. analysed the data. K.J.Y., M.N.R.A., C.S.H., H.B.D., G.R. W. and X.M.Y. discussed the experimental results. K.J.Y., C.S.H. and M.N.R.A. prepared the manuscript.

## Competing interests

The authors declare no competing interests.
