## [Peer Review File · Nature Communications]

REVIEWER COMMENTS

Reviewer #1 (Remarks to the Author):

This paper provides a wealth of state-to-state photodissociation data of H₂S at wavelength in the range of 138.99-139.13 nm.

Product distributions are observed for several absorption features in this range. Laser polarization dependence provides additional information. Seven reaction channels are identified (Fig 4), providing clues on the photodissociation mechanisms.

Astrochemical models currently use models of the photodissociation of H₂S that are too simplistic. Better understanding of the photodissociation of H₂S allows improved astrochemical modeling that may solve outstanding questions on sulfur abundances in astronomy.

This work by no means provides all the information needed by astrochemist. The paper mentions the need for high level ab initio calculations to complement the present experimental observations.

This work encourages further theoretical study and it provides ample data for benchmarking theoretical results.

Last year this group also published results on this system,

[Nat. Comm, 11, 1547, ref. 32 of the manuscript]

and they reported photolysis data at many wave length in the 122 - 155 nm range. In this new study, they achieve very

high resolution and report results for individual predissociating rotational levels in the 1B1 excited state of H₂S. The populations of the initial states were varied by controlling the beam temperature. The photodissociation product distributions depend strongly on the initial rotational state.

I believe the data provided by this study constitutes a substantial advance in the unraveling of the photodissociation mechanisms of H₂S.

Figure 3b i) shows the anisotropy of the photofragments.

Would it be possible to extract beta-parameters? If so, it would put a constraint on theoretical models.

I recommend publication of this work. Since there is a huge amount of data in this paper it may be somewhat overwhelming for readers not already familiar with this system.

I have some suggestions for the authors to consider.

Table 1 in the introduction gives an overview of the reaction channels, with the threshold energies are given in wavenumbers.

The channels are not sorted by energy. The absorption spectrum in figure 1 in the supplement is shown on a wavelength/nm scale.

Only in the last figure of the main text, figure 4, a diagram with the reaction pathways is shown, and the last figure of the supplement shows a potential energy surface. I understand the logic of this: first the experimental data is presented in detail

before the general conclusions are given.

My suggestion would be to add an overview figure in the introduction: an energy diagram that shows, e.g., in the middle the (X), the (1B1), and perhaps other states of H₂S in the middle, possibly with the absorption spectrum of Figure 1 of the supplement rotated over 90 degrees to present it on the energy scale. Then, on, e.g., the left side of this, on the same energy scale, the potential curve of H₂, shifted to show the dissociation to the H+H+S(3P₂) dissociation limit and another curve that shows the H₂ potential correlating with the H+H+S(1D₂) limit. The other side of the figure could show the SH(X) and the SH(A) curves. The vibrational product distribution could be shown by adding vibrational levels to the diatomic curves. Perhaps the suggested reaction pathways could be included schematically.

Another suggestion is to label the four absorption features in figure 2 of the supplement, and re-use these labels in the other figures. In the present manuscript the features are sometimes identified by wavenumber, sometimes by wavelengths, and also by rotational transitions.

Finally, I believe it may be helpful to have in the beginning a table with at least the X-state rotational levels identified by their rotational energy, rotational quantum numbers and parity. Also a column with the populations of the levels for T=3K and for T=15 K would be helpful.

Two minor comments: on line 228, the abbreviation HRA-PTS not defined. In figure 10 of the supplement I suggest labeling γ -axis in steps of 45 degrees, rather than 50, so that it is easier to see that the conical intersection occurs at the linear geometry.

Reviewer #2 (Remarks to the Author):

The present paper by Y. Zhao and colleagues reports on a comprehensive study of the photodissociation dynamics of H₂S in the vacuum-ultraviolet spectral range. Using photofragment excitation spectroscopy, REMPI spectroscopy, photofragment translational spectroscopy and imaging methods, a number of photofragmentation pathways has been identified and photofragment angular as well as product state distributions have been characterised. In the absence of suitable theoretical calculations on the excited states of H₂S, the results have mainly been discussed by drawing analogies to the well studied, similar case of H₂O photodissociation. In addition to the insights into fundamental photodissociation dynamics gained in this work, the present results are also of relevance to astrochemistry and will likely lead to a revision of models for H₂S photochemistry in the interstellar medium.

This is an impressive, if not heroic, study which was a thorough pleasure to read. The wealth of data presented, their meticulous analysis and the astonishing number of conclusions which the authors were able to draw in spite of the absence of detailed theoretical data render this study a landmark in photochemistry. This manuscript is certainly suitable for publication in Nat. Commun. subject to the clarification of a few minor points by the authors:

- The experiments use molecular beams with rather high seed ratios of H₂S, up to 30% in Ar. At these high concentrations, I would expect significant clustering of H₂S to occur in the beams and it is conceivable that a part of the detected photofragments in fact originates from clusters. Do the authors have any evidence for cluster formation, and how would cluster fragmentation impact their results and conclusions ?

- It strikes me that there is one big elephant in the room of this study: what exactly is this 1B₁ state which is initially photoexcited ? The authors mention that this is a Rydberg state, but which one ? In the text, the fragmentation dynamics originating from this state is discussed essentially in terms of couplings to lower valence states of H₂S. However, looking at the dense spectrum in Suppl. Fig. 1, I

would suspect a number of additional (Rydberg) states to be in the same region, possibly (likely ...) interacting with the doorway state and contributing to the dynamics. In other words, the photofragmentation may be even more complex than presently assumed. The authors should comment on these points.

- For the readers uninitiated to the electronic structure of H₂S, it is challenging to keep track of the various states and their couplings discussed in the text. Some sort of level scheme, either in the main text or the SI, which also highlights the couplings would considerably help to improve clarity.

Reviewer #3 (Remarks to the Author):

I will not have time to carefully go through the spectroscopy in this manuscript, so I will leave the detailed comments to another referee. What I will say is that the spectroscopy looks to be first-rate.

My only criticism is that so little effort was made to discuss the broader implications of the results obtained. The astrochemical importance of H₂S was established nicely in the intro paragraphs, but was not returned to in the discussion or concluding paragraphs.

As it stands now, I would argue that this should be a JCP spectroscopy paper. I would ask the authors to expand the scope of their conclusions.

Response to reviewers' comments

Reviewer #1:

This paper provides a wealth of state-to-state photodissociation data of H₂S at wavelength in the range of 138.99-139.13 nm. Product distributions are observed for several absorption features in this range. Laser polarization dependence provides additional information. Seven reaction channels are identified (Fig 4), providing clues on the photodissociation mechanisms.

Astrochemical models currently use models of the photodissociation of H₂S that are too simplistic. Better understanding of the photodissociation of H₂S allows improved astrochemical modeling that may solve outstanding questions on sulfur abundances in astronomy.

This work by no means provides all the information needed by astrochemist. The paper mentions the need for high level ab initio calculations to complement the present experimental observations. This work encourages further theoretical study and it provides ample data for benchmarking theoretical results.

Last year this group also published results on this system, [Nat. Comm, 11, 1547, ref. 32 of the manuscript] and they reported photolysis data at many wavelength in the 122 - 155 nm range. In this new study, they achieve very high resolution and report results for individual predissociating rotational levels in the ¹B₁ excited state of H₂S. The populations of the initial states were varied by controlling the beam temperature. The photodissociation product distributions depend strongly on the initial rotational state.

I believe the data provided by this study constitutes a substantial advance in the unraveling of the photodissociation mechanisms of H₂S.

Figure 3b i) shows the anisotropy of the photofragments. Would it be possible to extract beta-parameters? If so, it would put a constraint on theoretical models.

Author reply: Thank you for your comments. We considered this when preparing the original manuscript but decided it would not be helpful. Looking at fig 3b i), it's fairly clear that the outer rings show mildly perpendicular recoil anisotropy, whereas some of the inner rings indicate preferential parallel recoil. This conclusion we consider to be robust. But, as Supplementary Fig. 9 shows (admittedly for a neighbouring blended line), the precise form of the $P(v)$ and $\beta(v)$ distributions one will measure will depend on many parameters, e.g. the precise excitation wavelength (and the bandwidth of the excitation radiation) and the rotational temperature of the sample. On balance, therefore, we see little benefit in reporting best-fit β values from fitting the data measured in the present study; changing the experimental conditions would likely yield different β values.

I recommend publication of this work. Since there is a huge amount of data in this paper it may be somewhat overwhelming for readers not already familiar with this system. I have some suggestions for the authors to consider.

Table 1 in the introduction gives an overview of the reaction channels, with the threshold energies are given in wavenumbers. The channels are not sorted by energy. The absorption spectrum in figure 1 in the supplement is shown on a wavelength/nm scale. Only in the last figure of the main text, figure 4, a diagram with the reaction pathways is shown, and the last figure of the supplement shows a potential energy surface. I understand the logic of this: first the experimental data is presented in detail before the general conclusions are given.

My suggestion would be to add an overview figure in the introduction: an energy diagram that shows, e.g., in the middle the (X), the (¹B₁), and perhaps other states of H₂S in the middle, possibly with the absorption spectrum of Figure 1 of the supplement rotated over 90 degrees to present it on the energy scale. Then, on, e.g., the left side of this, on the same energy scale, the potential curve of H₂, shifted to show the dissociation to the H+H+S(³P₂) dissociation limit and another curve that shows the H₂ potential correlating with the H+H+S(¹D₂) limit. The other side of the figure could show the SH(X) and the SH(A) curves. The

vibrational product distribution could be shown by adding vibrational levels to the diatomic curves. Perhaps the suggested reaction pathways could be included schematically.

Author reply: Thank you for your comments. This is a very good suggestion, which we are happy to accommodate. The revised manuscript contains a new Figure 1, which includes many of these suggestions and is now specifically noted in the Introduction and cross-referred back to at relevant points in the subsequent narrative.

Another suggestion is to label the four absorption features in figure 2 of the supplement and re-use these labels in the other figures. In the present manuscript the features are sometimes identified by wavenumber, sometimes by wavelengths, and also by rotational transitions.

Author reply: Thank you for your comments. We also considered this when drafting the original manuscript but found using proxy labels to identify the peaks unhelpful – the quantum numbers and often the wavelength/wavenumber still had to be repeated to elaborate the point of interest. But readers would certainly be helped by including both wavenumber and wavelength scales in Fig. 2 and Supplementary Fig. 2 – both of which have been amended accordingly.

Finally, I believe it may be helpful to have in the beginning a table with at least the X-state rotational levels identified by their rotational energy, rotational quantum numbers and parity. Also a column with the populations of the levels for T=3K and for T=15 K would be helpful.

Author reply: Thank you for your comments. This too is a good suggestion as the process condition dependent parent level populations are not intuitively obvious, given the very different rotational and nuclear spin temperatures. Supplementary Table 1 has been revised to show these populations.

Two minor comments: on line 228, the abbreviation HRA-PTS not defined. In figure 10 of the supplement I suggest labeling y-axis in steps of 45 degrees, rather than 50, so that it is easier to see that the conical intersection occurs at the linear geometry.

Author reply: We thank the reviewer for highlighting these details, both of which we have acted on.

Reviewer #2:

The present paper by Y. Zhao and colleagues reports on a comprehensive study of the photodissociation dynamics of H₂S in the vacuum-ultraviolet spectral range. Using photofragment excitation spectroscopy, REMPI spectroscopy, photofragment translational spectroscopy and imaging methods, a number of photofragmentation pathways has been identified and photofragment angular as well as product state distributions have been characterised. In the absence of suitable theoretical calculations on the excited states of H₂S, the results have mainly been discussed by drawing analogies to the well studied, similar case of H₂O photodissociation. In addition to the insights into fundamental photodissociation dynamics gained in this work, the present results are also of relevance to astrochemistry and will likely lead to a revision of models for H₂S photochemistry in the interstellar medium.

This is an impressive, if not heroic, study which was a thorough pleasure to read. The wealth of data presented, their meticulous analysis and the astonishing number of conclusions which the authors were able to draw in spite of the absence of detailed theoretical data render this study a landmark in photochemistry. This manuscript is certainly suitable for publication in Nat. Commun. subject to the clarification of a few minor points by the authors:

- The experiments use molecular beams with rather high seed ratios of H₂S, up to 30% in Ar. At these high concentrations, I would expect significant clustering of H₂S to occur in the beams and it is conceivable that a part of the detected photofragments in fact originates from clusters. Do the authors have any evidence for cluster formation, and how would cluster fragmentation impact their results and conclusions?

Author reply: Thank you for your comments. We are confident that clusters (if present) are not significantly affecting the reported results. The PHOFEX spectra (Supplementary Fig. 2) show no variations with H₂S fraction (*e.g.* additional peaks or increased background signal level) other than what can be explained in terms of different parent monomer rotational temperatures. Below, we show $P(E_T)$ spectra derived from H atom TOF spectra measured following photodissociation of jet-cooled 30% H₂S in Ar (left panel) and 1% H₂S in Ar (right panel) samples at $\lambda = 139.051$ nm. Unsurprisingly, the signal-to-noise is better in the spectrum recorded using the richer mixture (more particles to detect), but the resolution is a little poorer (the spread of rotational energies in the precursor parent molecule is somewhat greater). Again, the difference in SH(A)/SH(X) ratio is fully interpretable in terms of the different parent rotational temperatures. We observed no signal that can be attributed to clusters of H₂S.

- It strikes me that there is one big elephant in the room of this study: what exactly is this ¹B₁ state which is initially photoexcited? The authors mention that this is a Rydberg state, but which one? In the text, the fragmentation dynamics originating from this state is discussed essentially in terms of couplings to lower valence states of H₂S. However, looking at the dense spectrum in Suppl. Fig. 1, I would suspect a number of additional (Rydberg) states to be in the same region, possibly (likely ...) interacting with the doorway state and contributing to the dynamics. In other words, the photofragmentation may be even more complex than presently assumed. The authors should comment on these points.

Author reply: Thank you for your comments. The ¹B₁ state in question has a near-integer quantum defect, so could (in principle) be attributed to an excitation from the HOMO (essentially a pure S(3p) orbital, with b₁ symmetry) to a 4s_{a1} orbital (with quantum defect $\delta \approx 1$) or to a 3d_{a1} orbital (with $\delta \approx 0$). Oscillator strength arguments encouraged Masuko et al (ref 25) and others to favour the 3d_{a1} ← b₁ assignment and this detail has been added at an appropriate point in the text.

- For the readers uninitiated to the electronic structure of H₂S, it is challenging to keep track of the various states and their couplings discussed in the text. Some sort of level scheme, either in the main text or the SI, which also highlights the couplings would considerably help to improve clarity.

Author reply: Thank you for your comments. This comment reinforces the suggestion from Reviewer 1 regarding the benefits of the new Figure 1 (see above).

Reviewer #3:

I will not have time to carefully go through the spectroscopy in this manuscript, so I will leave the detailed comments to another referee. What I will say is that the spectroscopy looks to be first-rate.

My only criticism is that so little effort was made to discuss the broader implications of the results obtained. The astrochemical importance of H₂S was established nicely in the intro paragraphs but was not returned to in the discussion or concluding paragraphs.

As it stands now, I would argue that this should be a JCP spectroscopy paper. I would ask the authors to expand the scope of their conclusions.

Author reply: Thank you for your comments. The existing conclusion highlights H₂S photolysis as a potentially important source of S atoms observed in a range of interstellar environments and highlights the exquisite detail that will be required in order to gain a much more detailed picture of the SH/H₂S number density ratio induced by the interstellar radiation field. We have returned to the astrochemical importance of this work in the last paragraph of the main text (Page 15).

We very much hope that you judge these responses to be appropriate and hope to see the manuscript published in a future issue of *Nature Communications*.

REVIEWERS' COMMENTS

Reviewer #1 (Remarks to the Author):

The authors addressed all my comments and questions. I believe they succeeded in making the paper more accessible. I very much agree with Reviewer 2 who describes this work as a "landmark in photochemistry" and also with the assessment of Reviewer 3 that the spectroscopy is "first-rate". Although this work would certainly also make a great JCP paper, I believe it deserves the perhaps wider attention it may receive as publication in Nature Communications.

Reviewer #2 (Remarks to the Author):

The authors have satisfactorily addressed my comments. I congratulate them on their results and can recommend this paper for publication in Nat. Commun.

Response to reviewers' comments

Reviewer #1:

The authors addressed all my comments and questions. I believe they succeeded in making the paper more accessible. I very much agree with Reviewer 2 who describes this work as a "landmark in photochemistry" and also with the assessment of Reviewer 3 that the spectroscopy is "first-rate". Although this work would certainly also make a great JCP paper, I believe it deserves the perhaps wider attention it may receive as publication in Nature Communications.

Author reply: Thank you very much for your comments.

Reviewer #2:

The authors have satisfactorily addressed my comments. I congratulate them on their results and can recommend this paper for publication in Nat. Commun.

Author reply: Thank you very much for your comments.

We thank the reviewers for the hard work in our paper peer review process and many constructive suggestions!